# Immature Teratoma of the Ovary—A Narrative Review

**DOI:** 10.3390/cancers17183041

**Published:** 2025-09-18

**Authors:** Giuseppe Marino, Serena Negri, Filippo Testa, Jasmine Corti, Daniela Giuliani, Daniele Lugotti, Tommaso Grassi, Marta Jaconi, Alessandra Casiraghi, Cristina Maria Bonazzi, Robert Fruscio

**Affiliations:** 1Obstetrics and Gynecology, Department of Medicine and Surgery, University of Milan-Bicocca, 20126 Milan, Italy; g.marino38@campus.unimib.it (G.M.); s.negri20@campus.unimib.it (S.N.); d.lugotti@campus.unimib.it (D.L.); 2UO Gynecology, Fondazione IRCCS San Gerardo dei Tintori, 20900 Monza, Italy; daniela.giuliani2@gmail.com (D.G.);; 3UO Pathology, Fondazione IRCCS San Gerardo dei Tintori, 20900 Monza, Italy; 4UO Radiology, Fondazione IRCCS San Gerardo dei Tintori, 20900 Monza, Italy

**Keywords:** ovarian cancer, malignant ovarian germ cell tumors, immature teratomas, fertility-sparing, management, controversies

## Abstract

Immature teratomas are rare ovarian tumors, predominantly affecting adolescents and young women. Surgery—preferably with a fertility-sparing approach—remains the cornerstone of treatment for both early and advanced-stage disease, potentially followed by adjuvant chemotherapy. Fertility outcomes after fertility-sparing surgery are favorable, with high pregnancy and live birth rates. However, the rarity of the disease contributes to the limited availability of data in the literature. A comprehensive review of immature teratomas is still lacking, highlighting the need for consolidated information on their management and outcomes.

## 1. Introduction

Ovarian cancer is currently the eighth most common cancer and the eighth leading cause of cancer-related death among women worldwide [1]. However, the term “ovarian cancer” encompasses a heterogeneous group of tumors, with significant variability depending on the specific histological subtype.

Ovarian tumors can be classified based on their origin into the following main categories [2,3]: •**Epithelial tumors** are believed to arise from the coelomic epithelium lining the ovary. However, recent studies suggest that many high-grade serous carcinomas, a common type of epithelial ovarian cancer, may actually originate in the fimbrial end of the fallopian tube [4]. Moreover, the origin of certain ovarian cancer histotypes is often associated with pre-existing conditions such as endometriosis, or may result from the progression or malignant transformation of benign or borderline ovarian cysts.•**Sex cord-stromal tumors** originate from the ovarian stroma and sex cords.•**Germ cell tumors** are derived from primordial germ cells.

Epithelial tumors are the most common, accounting for approximately 90% of all ovarian neoplasms. Among these, the high-grade serous subtype represents more than two-thirds of cases [3].

Malignant ovarian germ cell tumors (MOGCTs) account for only 2–3% of all malignant ovarian tumors. Notably, 58% of MOGCTs occur in patients under the age of 20, and 82% are diagnosed between ages 14 and 54, predominantly affecting women of reproductive age [5]. Recent epidemiological data show a stable incidence rate of approximately 0.4 cases per 100,000 inhabitants per year in Europe [5,6].

The various histological subtypes of MOGCTs reflect the developmental stage of the germ cell at the time of neoplastic transformation, ranging from undifferentiated germ cells to fully differentiated adult tissues [7]. According to the World Health Organization (WHO) classification, MOGCTs are categorized into several distinct histological types, as summarized in Table 1.

Immature teratomas (ITs) are the most common non-dysgerminomatous subtype of MOGCTs, accounting for approximately one-third of cases, with a prevalence of 1.5 per 100,000 women [6], and predominantly affect individuals between 15 and 30 years of age [5]. Despite their relatively high prevalence within the MOGCT group and the clinical importance due to their occurrence in young women, there is a paucity of specific data in the literature regarding this unique entity. Published studies often aggregate data from heterogeneous MOGCT populations, and cases of pure ITs are relatively rare, which complicates evidence-based decision-making for individual patients. Previous reviews on the topic are present in the literature, but only on specific issues, and they do not offer a comprehensive vision focusing on all the aspects regarding ITs [8,9]. The aim of this review is to summarize the current evidence on ovarian ITs, starting from epidemiology and diagnosis to management and follow-up, including fertility-sparing treatment, providing guidance for this rare disease.

## 2. Methods

This narrative review was conducted to provide a comprehensive and updated overview of ovarian ITs, paying particular attention to the diagnostic work-up, the prognosis, the management in terms of surgical and medical management, the fertility outcomes after a fertility-sparing approach, and follow-up strategies. A literature search was carried out using PubMed/MEDLINE, Scopus, and Web of Science, covering studies published in English from January 1980 to June 2025. The following search terms were used in various combinations: “immature teratoma”, “ovarian germ cell tumor”, “fertility preservation”, “surgical management”, “chemotherapy”, “growing teratoma syndrome”, and “follow-up”. Boolean operators (AND, OR) were applied to optimize sensitivity and specificity. Articles were selected based on clinical relevance and included original research, systematic and narrative reviews, clinical guidelines, and expert opinions. Studies involving mixed histologies were excluded to give a better focus on ITs. Full texts, abstracts, and main manuscripts were assessed for eligibility. Reference lists of key articles were also manually searched to identify additional relevant sources.

Being a narrative review, no formal methodological quality assessment or meta-analysis was performed. The selected literature was synthesized descriptively and organized into a subsection and multiple subsubsections to highlight consistent findings, evolving practices, and knowledge gaps.

## 3. Immature Teratomas

### 3.1. Diagnosis

Unlike mature teratomas (which are benign cysts), ITs contain embryonic tissue, the presence and degree of which correlate directly with prognosis. Clinical presentation may vary: although the diagnosis could be totally accidental during a routine ultrasound, most patients report abdominal pain and the presence of an abdominal mass, which can rapidly grow, reflecting the tumor’s aggressive growth rate. In approximately 10% of cases, presentation may include an acute abdomen due to rupture, hemorrhage, or torsion [10,11].

The diagnostic process involves several key components:

**Tumor markers** [11,12]: In the presence of an ovarian mass of uncertain nature, tumor markers play a crucial role not only in aiding diagnosis but also in monitoring therapeutic response and facilitating post-treatment surveillance for early detection of recurrence. Baseline serum levels of Cancer Antigen 125 (CA-125), alpha-fetoprotein (AFP), beta subunit of human chorionic gonadotropin (β-hCG), lactate dehydrogenase (LDH), and inhibin B should be assessed prior to surgery or chemotherapy.

Among these, AFP is the only marker that typically shows alterations in ITs of the ovary, although levels may remain within the normal range in up to 50% of cases (Table 2). Elevated AFP levels may also suggest the presence of yolk sac tumor components in the context of a mixed germ cell tumor [13].

**Table 2 cancers-17-03041-t002:** Tumor markers in Immature teratoma of the ovary.

	AFP	bHCG	LDH
Immature teratoma	+/-	-	-

**Ultrasound** [14,15]: On ultrasound, ITs typically appear as large, unilateral ovarian cysts with a solid–solid multilocular pattern (according to International Ovarian Tumor Analysis (IOTA) classification). The solid components exhibit atypical echogenicity and are generally poorly vascularized (color score 1 or 2). The solid areas often contain calcifications and irregular shadowing, producing two characteristic patterns: a foam pattern—with a frothy appearance, and a tangled string pattern—resembling intertwined laces.

**Second-line imaging techniques**: abdomen–chest computed tomography (CT) scan, magnetic resonance imaging (MRI), and positron emission tomography (PET) [15,16]. Second-line imaging techniques, such as abdominal–thoracic CT scan, abdomino-pelvic MRI, and total body PET, are valuable for diagnosis, staging, and post-treatment monitoring. On CT scan, solid components of ITs typically appear large and irregular, with coarse calcifications, small foci of fat, and areas of hemorrhage, features characteristic of this tumor type.

**Histology**: Histopathological analysis remains the gold standard for definitive diagnosis. Macroscopically, the mass is usually large (mean diameter ~16 cm), unilateral, and exhibits both cystic and solid components interspersed with hemorrhagic and necrotic areas [17,18]. Microscopically, ITs show immature elements derived from two or all three embryonic germ layers (ectoderm, mesoderm, and endoderm), containing both mature and immature tissues; most notably, immature neuroepithelium, which is key for both diagnosis and grading [3]. The neuroepithelial component may appear sarcomatoid or arranged into rosettes, pseudorosettes, or primitive tubules (Figure 1) [17]. Foci of cartilage and fetal muscle may also be observed, although they are not diagnostic of immaturity. In some cases, elements of other germ cell tumors, such as yolk sac tumor or embryonal carcinoma, may be present, classifying the neoplasm as a mixed germ cell tumor (Figure 2a,b). However, small foci (<3 mm) do not appear to significantly impact prognosis [18], and WHO guidelines have not yet provided specific recommendations for distinguishing between ITs and mixed germ cell tumors in these cases.

Frozen section or biopsy may be employed to guide intraoperative decision-making in cases of highly suspicious lesions. However, extensive sampling is essential to detect focal areas of immature neuroepithelium [19,20].

**Immunochemistry, genetic and molecular aspects:** Molecular and genetic studies play an increasingly important role in the diagnosis and differential diagnosis of immature ovarian teratomas. These tumors are typically diploid or near-diploid and lack the chromosomal abnormalities commonly found in other malignant germ cell tumors. For example, pure ITs do not show isochromosome 12p [i(12p)] or chromosome 12p amplification when evaluated using fluorescence in situ hybridization (FISH). This distinguishes them from many immature and non-immature components of mixed germ cell tumors and supports the hypothesis of a distinct origin for pure ITs [21]. ITs are also generally negative at the immunochemistry evaluation for AFP, OCT4, CD30, and PLAP, which are markers more typical of other germ cell tumor components. Molecular profiling also helps exclude mixed germ cell tumors or somatic-type malignancies (e.g., primitive neuroectodermal tumor (PNET)) arising within a teratoma. In addition to molecular and cytogenetic findings, ancillary immunohistochemical markers such as SALL4 and Glypican-3 offer important diagnostic insights, particularly in the differential diagnosis of immature ovarian teratomas. While SALL4 is a highly sensitive marker for germ cell tumors and is frequently expressed in both immature teratomas and yolk sac tumors, it lacks specificity and should be interpreted cautiously in isolation [22]. Glypican-3, on the contrary, tends to be strongly and diffusely positive in yolk sac tumors, but is typically negative or only focally positive in ITs, offering helpful discriminatory value [23]. In conclusion, due to overlapping immunophenotypic expression among germ cell tumors, no single marker is definitive, and immunohistochemistry should always be interpreted in conjunction with histological and molecular data for accurate diagnosis and tumor classification.

### 3.2. Prognostic Factors

**Age** appears to play a significant role in the risk of recurrence. Recurrence rates are very low in the pediatric population, but become more relevant in individuals over the age of 30. In a large study including more than one thousand patients with ITs of the ovary, Chan et al. highlighted the impact of age on prognosis, reporting a disease-specific survival of 97% in patients under 18 years compared to 89% in those over 18 [24], even if no differences were found considering only stage I disease. In 2021, Guo and colleagues [25] presented a comprehensive study using the American SEER (Surveillance, Epidemiology, and End Results) database, covering patients diagnosed in the United States between 1973 and 2012. Among more than 1900 patients with MOGCTs, 820 had ITs. This study confirmed the prognostic significance of age, reporting hazard ratios for overall survival of 2.2 for patients over 30 years and 23.7 for those over 50. These findings are also reflected in therapeutic strategies: in pediatric patients with stage I disease, observation after surgery is typically sufficient, whereas in adults, the role of adjuvant chemotherapy remains controversial, except in stage IA grade 1 tumors [26,27,28].

Chan et al. [24] also identified **ethnicity** as a factor influencing disease incidence, though not directly affecting prognosis. The incidence of ITs was found to be approximately three times higher in individuals of Asian descent compared to Caucasians and African Americans. However, these data must be interpreted cautiously, as the studied population was predominantly of Asian origin but born and raised in the U.S., thus excluding environmental factors. The same study reported worse outcomes in African American patients; however, this may be influenced by disparities in access to healthcare, which continue to affect Black women in the United States and could account for the observed differences.

**Histological grade** of the tumor is considered the primary prognostic factor for recurrence, regardless of patient age. Tumor grade is determined exclusively based on the amount of immature neuroepithelium present. The original grading system, proposed by Norris in 1976, categorizes tumors into three grades based on the amount of neuroepithelium observed under low-power magnification [29]:•Grade 1: <1 low power field (using a 4× objective and a 10× lens, 40× of magnification) per slide;•Grade 2: ≥1 but <3 low power fields in any slide;•Grade 3: ≥3 low-power fields in any slide.

A more recent classification by O’Connor and Norris [18] groups grades 2 and 3 as high-grade disease and grade 1 as low-grade, improving reproducibility in pathological reporting. In a study published in Cancer, involving 98 pediatric and 81 adult patients, no recurrences were observed in grade 1 tumors. In contrast, recurrence rates in higher stages were 21% and 20% in pediatric and adult patients, respectively [30].

**Stage** is another independent prognostic factor. The International Federation of Gynecology and Obstetrics (FIGO) staging system used for ITs is the same as that for epithelial ovarian cancer [31]. While stages I and II have an excellent prognosis (>95%), this drops considerably for advanced stages (81% and 71% for stage III and IV, respectively) [24,32,33,34,35].

**The presence of yolk sac tumor components** also negatively impacts prognosis. As noted earlier, foci of yolk sac tumor exceeding 3 mm are associated with a higher risk of recurrence and worse outcomes [36].

Interestingly, approximately 25% of ITs are associated with peritoneal implants composed of mature tissues, termed grade 0 implants. This condition, known as gliomatosis peritonei, appears to result not from low-grade metastasis, but rather from metaplasia of submesothelial peritoneal cells in response to tumor-derived growth factors [37]. The only way to distinguish gliomatosis peritonei from peritoneal metastases is through histological biopsy. Although gliomatosis may be associated with a slightly increased risk of recurrence, it does not appear to affect overall survival. For this reason, it does not constitute an indication for chemotherapy [37,38,39]. In rare cases, nodal gliomatosis (the presence of mature glial elements in lymph nodes) can be observed. Even in these cases, systematic lymphadenectomy is not recommended unless lymph nodes appear clinically or radiologically suspicious for metastatic involvement.

A very rare complication of immature teratoma, but still described especially in young patients, is Anti-N-methyl-D-aspartate (NMDA) receptor encephalitis, a severe autoimmune neurological disorder characterized by antibodies against NMDA-type glutamate receptors. Firstly, described and strongly associated with mature teratoma [40], this paraneoplastic condition could also be associated with the immature counterpart. In fact, ITs, containing abundant neuroectodermal elements, are thought to express NMDA receptor subunits ectopically, potentially initiating an autoimmune response [37]. The resulting encephalitis presents with psychiatric symptoms, seizures, memory deficits, and autonomic dysfunction [41]. Notably, neurological symptoms often precede tumor detection, emphasizing the need for prompt oncological evaluation in suspected cases, also because tumor removal is the standard and consolidated treatment for the condition [42].

### 3.3. Management

#### 3.3.1. Management of Stage I Disease

The survival rate after a diagnosis of a stage I IT of the ovary remains excellent, close to 100% as reported by many authors and summarized in Table 3.

Surgery remains the cornerstone of treatment for ITs of the ovary, as emphasized in the guidelines of both the European Society for Medical Oncology (ESMO) and the American National Comprehensive Cancer Network (NCCN) [56,57]. Given that this disease predominantly affects young women, often before childbearing desires have been fulfilled or even expressed, fertility-sparing surgery should be prioritized whenever feasible [56,57]. For stage I disease, the standard surgical approach consists of a unilateral salpingo-oophorectomy accompanied by comprehensive surgical staging, including omental biopsy, multiple peritoneal biopsies, peritoneal washing cytology, and thorough exploration of the abdominal cavity [32,58]. Biopsy of the contralateral ovary should be avoided due to the risk of damage and impairment of fertility, unless there is clinical or radiological suspicion of metastatic involvement. Adequate surgical staging is critical for proper disease classification and informs subsequent management, while significantly reducing the risk of recurrence [56,58].

Similar to testicular germ cell tumors in males, which share biological and clinical features with MOGCTs, lymphadenectomy has not been shown to improve progression-free or overall survival and is instead associated with increased short- and long-term morbidity [59]. Therefore, systematic lymphadenectomy is not recommended. Selective removal of lymph nodes should only be considered when there is radiological or intraoperative suspicion of nodal involvement.

Laparotomy is often the preferred surgical route due to the large size of these tumors and the associated risk of upstaging if intra-abdominal rupture or morcellation occurs. However, minimally invasive approaches (laparoscopic or robotic) may be considered in selected cases, particularly in high-volume oncologic centers, without compromising oncologic outcomes [56,57]. Similarly, although unilateral adnexectomy remains the standard approach, cystectomy with preservation of healthy ovarian tissue may be considered. Small case series have shown no significant difference in oncologic outcomes between the two approaches [60,61]. Preserving ovarian tissue could be crucial for fertility, especially considering not only a relapse on the remnant ovary, but also that 10–15% of patients may develop a mature teratoma in the contralateral ovary, either at diagnosis or subsequently, which may require further surgical intervention and increase the risk of oocyte loss [46,61].

Following surgery, there is no strong consensus on the optimal strategy for adjuvant chemotherapy or surveillance in patients with stage I disease.

For stage IA, grade 1 ITs, surveillance alone is widely accepted, based on historical studies showing survival rates close to 100% with surgery alone [62]. However, for stage IA (grades 2–3) and stages IB/IC (any grade), the post-surgical approach remains a topic of debate. An increasing number of Authors now advocate for active surveillance in all stage I cases, reserving adjuvant chemotherapy for relapses [26,44]. This paradigm shift is inspired in part by the management of stage I testicular germ cell tumors, which have largely moved away from adjuvant chemotherapy in favor of surveillance [63]. Additionally, pediatric trials have demonstrated no significant difference in outcomes between patients who received adjuvant chemotherapy and those managed with surveillance [64,65].

Although the bleomycin/etoposide/cisplatin (BEP) regimen is standard for MOGCTs and has not shown major long-term effects on fertility, it is nonetheless associated with acute and late toxicities, including secondary malignancies [66,67,68,69,70,71,72], as reported in Table 4.

A 2010 multicenter Italian study by the Multicentre Italian Trial in Ovarian Cancer (MITO) group observed excellent long-term outcomes with surveillance alone in 28 patients with stage I ITs, supporting the selective use of chemotherapy only in cases of recurrence [44]. This finding was confirmed in a larger retrospective study by Bergamini et al. [26], which analyzed 108 patients with stage I ITs treated with fertility-sparing surgery across the Charing Cross Hospital in London and MITO centers in Italy. The cohort included 66 stage IA, 3 stage IB, and 39 stage IC patients.

Adjuvant chemotherapy was administered in 25% of cases, while the remaining 75% underwent surveillance alone. The recurrence rate was 7.4% (2/25) in the chemotherapy group and 11.1% (9/81) in the surveillance group, a non-significant difference. All relapsed patients were successfully treated with chemotherapy. These findings support the use of surveillance in adult patients with stage I ITs, regardless of grade, with chemotherapy reserved for recurrences [26]. In contrast, a meta-analysis by Li et al. reported worse disease-specific survival in adult patients who did not receive adjuvant chemotherapy [8]. Before initiating a surveillance-only strategy, accurate staging and histological diagnosis must be ensured. Incomplete staging has been associated with significantly higher recurrence rates [26,36,73], highlighting the importance of centralizing care in high-volume centers with expertise in germ cell tumors. Moreover, thorough counseling is essential to communicate the risks and benefits of both surveillance and chemotherapy. Finally, there is a clear need for prospective clinical data, despite the inherent challenges due to the rarity of this disease. In response, the MITO group has launched a prospective registry, named MITO-9b, to address the outcomes in this population [27].

#### 3.3.2. Management in Advanced Stages (II–IV)

Although the majority of ITs are diagnosed at an early stage (60–70%), systemic spread is not uncommon and may lead to stage progression and a worse prognosis. The management of advanced-stage ITs remains a subject of ongoing debate. Data from the published literature on advanced-stage ITs are presented in Table 5, despite the challenges in extrapolating from reports involving MOGCTs.

Similar to epithelial ovarian cancer, some authors advocate for primary debulking surgery in advanced disease, arguing that removing all the macroscopic disease may reduce chemoresistant clusters and should therefore be considered as the initial approach, followed by adjuvant chemotherapy [75]. However, ITs are highly chemo-sensitive, and fertility preservation remains a crucial concern, even in advanced stages. For this reason, neoadjuvant chemotherapy (NACT) should always be considered when a complete debulking or a fertility-sparing surgery in patients who desire preserving fertility cannot be initially achieved [56,57]. Moreover, for stage IV disease or in patients unfit for surgery, systemic therapy as a first-line treatment may be beneficial, followed by a less radical surgical procedure. In this context, Talukdar et al. proposed NACT as a feasible strategy for patients with bulky disease. This approach can reduce tumor burden, enable fertility-preserving surgery, and has shown promising results in terms of both oncologic and reproductive outcomes [76]. However, no patients with advanced IT of the ovary were enrolled in the NACT group. Currently, a standard NACT regimen consists of 3 cycles of BEP, with the possibility of extending to 4–6 cycles to achieve better response rates [77]. Nevertheless, despite the rationale, this strategy is not yet considered the standard of care [78]. Primary debulking surgery remains widely used in this setting and does not appear to compromise progression-free survival or overall survival, as reported by our group [74] and by Nasioudis et al. [79]. However, the retrospective nature of these studies, the heterogeneity of ovarian germ cell tumor subtypes, and the small sample sizes represent major limitations to defining a standard of care, as outlined by Fumagalli et al. in their review of the literature [80]. When managing advanced or relapsed ITs, clinicians should be aware of an uncommon but characteristic phenomenon known as growing teratoma syndrome. This syndrome is characterized by a paradoxical increase in tumor size following neoadjuvant chemotherapy, despite normalization of tumor markers. It is thought to result from the maturation of immature tissue and necrosis, potentially leading to a substantial increase in tumor volume [81]. In such cases, especially in patients with large tumors at high risk of complications such as hemorrhage, rupture, or organ involvement (e.g., bowel or lung), it may be appropriate to initiate treatment with a low-dose induction chemotherapy regimen (etoposide 100 mg/m^2^ plus cisplatin 20 mg/m^2^ on days 1 and 2), repeated weekly until the patient is stable enough to receive standard therapy [82]. Two to four weeks after completion of NACT, imaging studies should be performed to assess resectability and to guide optimal surgical or non-surgical management.

#### 3.3.3. Management of Relapses

There is limited research on the management of recurrences following treatment for ITs. Overall, the recurrence rate is estimated at 14–20%, with the highest risk occurring within 24 months of initial treatment [34]. In most cases, recurrence presents as peritoneal spread, while retroperitoneal relapse is rare. In the evaluation of suspected recurrence, alongside clinical assessment and ultrasound, second-level imaging techniques such as pelvic-abdominal MRI, contrast-enhanced thoracoabdominal CT, and PET scans are essential to determine the extent and resectability of the disease. Surgical intervention is generally reserved for isolated recurrences or cases in which complete resection is achievable. According to the ESMO guidelines, for platinum-sensitive relapses (defined as disease progression occurring more than 4–6 weeks after completion of chemotherapy), a second-line regimen with ifosfamide and platinum (IP), with or without paclitaxel (TIP), is recommended [56]. An alternative option is the POMB/ACE regimen (cisplatin, vincristine, methotrexate, bleomycin/actinomycin D, cyclophosphamide, etoposide), which is considered to maintain a safe profile on fertility preservation [83]. For platinum-resistant disease, possible salvage regimens include vincristine/actinomycin D/cyclophosphamide (VAC) or paclitaxel/gemcitabine, although these options are associated with limited response rates (10–30%) [56].

#### 3.3.4. Fertility Outcomes

Only a limited number of studies have reported fertility outcomes in patients with both stage I and advanced-stage ITs. However, it is important to acknowledge that chemotherapy may have long-term adverse effects, including impaired ovarian function and reduced fertility [84,85]. The extent of fertility impairment depends on several factors, such as the type and duration of chemotherapy, patient age at diagnosis, presence of comorbidities, and extent of surgical intervention. Nonetheless, available data on fertility outcomes are generally reassuring. Most studies report fertility results in patients with MOGCTs as a whole, without specifying outcomes by histologic subtype, thus limiting the extrapolation of IT-specific data.

Low (2000) [86], Zanetta (2001) [77], Tangir (2003) [87], and Mangili (2011) [36] reported 95% (19/20), 87.5% (28/32), 76% (29/38), 80% (12/15) of patients with MOGCTs getting pregnant in patients who tried to conceive after FSS, respectively, but no specific data on ITs are possible to deduce. More specifically, Bonazzi et al. (1994) [43] reported that 5 out of 6 patients with IT of the ovary conceived after FSS. Tamauchi et al. [88] noted that 20 out of 42 patients with an initial diagnosis of IT had successful live births, although data on how many actively attempted pregnancy were not available. Our own group recently reported a large cohort of pure ovarian ITs, showing a pregnancy rate of 78.8% (37/47) among patients with only stage I disease who underwent FSS and subsequently attempted conception [89]. In a separate cohort of patients with advanced-stage ITs, 46% (6/13) of those desiring pregnancy were able to conceive following FSS according to our publication [74], indicating that favorable fertility outcomes may still be achievable even in more advanced disease. Despite the limited amount of IT-specific data, the available evidence supports the feasibility and safety of fertility preservation in this population, and justifies the effort to pursue fertility-sparing approaches whenever oncologically appropriate.

### 3.4. Follow-Up

The ESMO guidelines recommend follow-up schedules based on disease stage and whether adjuvant chemotherapy was administered [56]. In stage I patients treated with surgery alone, without adjuvant chemotherapy, a strategy of active and strict surveillance is recommended, following the protocol proposed by Vasquez et al. [90], as summarized in Table 6.

For patients who underwent adjuvant chemotherapy, follow-up visits should include physical and pelvic examinations, tumor markers evaluation every 3 months for the first 2 years, every 6 months during the third to five years, and then yearly until progression. The NCCN guidelines [57], incorporating recommendations by Salani et al. [91], suggest a surveillance approach for MOGCTs that closely resembles ESMO’s, though without adjustments based on adjuvant treatment. To date, no consensus exists on the optimal use or frequency of imaging modalities in the follow-up of IT patients. Based on our institutional experience, we recommend a transvaginal ultrasound every 3 months in patients who have undergone fertility-sparing surgery to detect possible recurrences in the contralateral ovary or pelvic structures. Additionally, second-level imaging, such as CT scans or abdominal MRI, should be performed at least annually, alternating when possible with abdominal ultrasound to reduce radiation exposure. The practice of laparoscopic second-look surgery, once advocated by the Charing Cross group [92], is no longer considered standard. However, in cases of strong clinical or radiologic suspicion of relapse, a second-look procedure with biopsy may still be warranted to confirm or exclude recurrence.

## 4. Conclusions

ITs are associated with an excellent prognosis, particularly when compared to epithelial ovarian cancers. However, their rarity, combined with their predominance in younger patients, contributes to a relative lack of robust data in the literature, which continues to challenge clinical decision-making. Additionally, controversies on the origin, molecular and pathological aspects, prognosis, and proper management are still present, both for ITs and for MOGCTs overall [93]. There is a clear need for greater standardization in the management of ITs, including efforts to centralize care in high-volume oncologic centers and to prioritize fertility preservation whenever oncologically appropriate. International collaboration, along with the prospective collection of clinical data, represents a crucial path forward in improving outcomes, especially in cases of high-risk disease and recurrence.

## Figures and Tables

**Figure 1 cancers-17-03041-f001:**
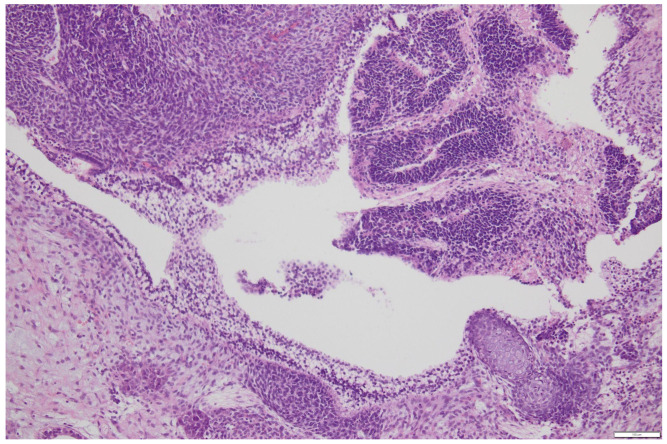
Neuroepithelium organization in rosettes in a primitive immature teratoma of the ovary-courtesy of Department of Pathology, Fondazione IRCCS San Gerardo dei Tintori, Monza, Italy. Scale bar: 100 μm.

**Figure 2 cancers-17-03041-f002:**
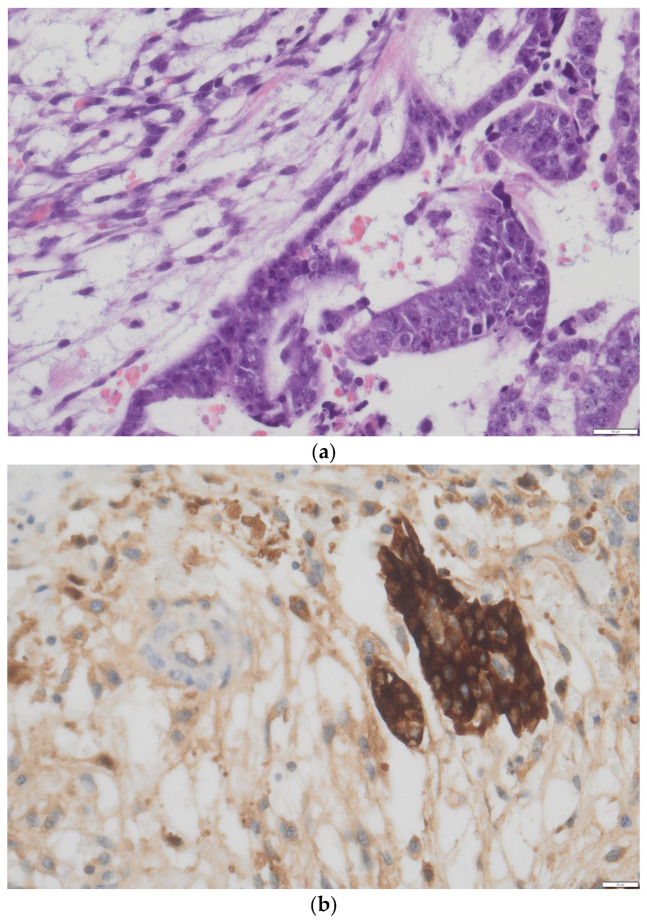
(**a**) Minute area of yolk sac tumor interspersed in immature teratoma component in a H&E slide, doubt for mixed germ cell tumor, courtesy of Department of Pathology, Fondazione IRCCS San Gerardo dei Tintori, Monza, Italy. (**b**) Confirmation obtained with immunohistochemistry (positive staining for AaFP) showing an area of yolk sac tumor interspersed in immature teratoma component, defining the neoplasia as mixed germ cell tumor, courtesy of Department of Pathology, Fondazione IRCCS San Gerardo dei Tintori, Monza, Italy. Scale bar: (**a**) 100 μm; (**b**) 20 μm.

**Table 1 cancers-17-03041-t001:** World Health Organization classification of MOGCTs.

**Dysgerminoma** **Non-dysgerminoma** • **Immature teratoma** ○ **Grade 1** ○ **Grade 2** ○ **Grade 3** • **Yolk sac tumor** • **Embryonal carcinoma** • **Non-gestational Choriocarcinoma** • **Mixed germ cell tumor**

**Table 3 cancers-17-03041-t003:** Clinical characteristics and outcomes of patients with stage I IT enrolled in studies published in the literature.

Author and Year of Publication	N Patients	Stage (IA/IB/IC)	Grade (G1/G2/G3)	FSS	Chemo	Relapses	Deaths
Bonazzi, 1994 [43]	26	18/0/8	7/13/6	26/26	6/26	2/26	0/26
Mangili, 2010 [44]	28	19/2/7	9/12/7	24/28	9/28	6/28	0/28
Vicus, 2011 [45]	32	27/0/5	13/11/8	29/32	5/32	4/32	3/32
Alwazzan, 2015 [46]	22	12/1/9	8/2/12	19/22	16/22	0/22	0/22
Pashankar, 2016 * [30]	43	502/147 (IB-IC)	-	-	43/43	1/43	1/43
Reddihalli, 2015 [47]	16	5/0/5	-	15/16	15/16	0/16	0/16
Jorge, 2016 [35]	756	-	-	-	379/756	-	13/756
Chan, 2016 [24]	418	-	103/114/118 (83 missing)	-	-	-	2/418
Mangili, 2017 [48]	49	22/2/12 (13 unknown)	-	-	15/49	11/49	-
Newton, 2019 * [49]	32	-	-	-	6/32	2/32	0/32
Bergamini, 2020 [26]	108	66/3/39	31/41/36	100/108	27/108	11/108	1/108
Wang, 2020 [50]	75	14/0/12 (49 unknown)	35/25/15	75/75	51/75	4/75	1/75
Mangili, 2021 [27]	24	16/1/7	9/2/13	24/24	5/24	1/24	0/24
Nasioudis, 2021 [51]	272	-	0/115/157	-	170/272	-	8/272
Graham, 2022 [52]	39	26/0/7 (6 undefined)	11/15/12	38/39	6/39	6/39	0/39
Zhang, 2022 [53]	32	6/0/26	12/20 (G2–G3)	32/32	16/32	2/32	0/32
Li, 2023 [54]	126	55/0/71	39/57/30	105/126	81/126	13/126	2/126
Marino, 2024 [55]	74	59/1/14	28/28/18	74/74	9/74	10/74	0/74

- Not reported or not possible to extrapolate from aggregate MOGCTs data. * Adult population.

**Table 4 cancers-17-03041-t004:** Toxicity related to the BEP schedule and long-term adverse effects [56].

Pulmonary toxicity (Bleomycin) Neuropathy (Platin) Nephrotoxicity (Platin) Raynaud’s phenomenon Cardiovascular impairment High tone hearing loss (Platin) Osteoporosis (if radical surgery) Infertility Acute myeloid leukemia (Etoposide)

**Table 5 cancers-17-03041-t005:** Clinical characteristics and outcomes of patients with advanced-stage ITs enrolled in studies published in the literature.

Author and Year of Publication	N Patients	Stage (II/III/IV)	Grade (G1/G2/G3)	FSS	Chemo	Relapses	Deaths
Bonazzi, 1994 [43]	5	2/3/0	2/2/1	4/5	3/5	3/5	0/5
Vicus, 2011 [45]	2	1/1/0	0/1/1	1/2	2/2	1/2	1/2
Alwazzan, 2015 [46]	5	3/2/0	-	-	4/5	1/5	0/5
Pashankar, 2016 * [30]	38	5/27/6	-	-	38/38	9/38	5/38
Reddihalli, 2015 [47]	5	0/5/0	-	-	5/5	4/5	1/5
Jorge, 2016 [35]	236	68/143/25	-	-	190/236	-	48/236
Chan, 2016 [24]	116	24/71/21	8/22/58 (28 missing)	-	-	-	21/116
Newton, 2019 * [49]	10	-	-	-	-	2/10	-
Marino, 2025 [74]	17	4/12/1	2/6/9	13/17	14/17	5/17	0/17

- Not reported or not possible to extrapolate from aggregate MOGCTs data. * Adult population.

**Table 6 cancers-17-03041-t006:** Active surveillance program based on ESMO guidelines for ITs (and in general MOGCTs) management.

Timing	Clinical Examination	Ultrasound	Tumor Markers	Chest Radiography	Total Body CT Scan
I year	Each month	Every 2 months	Every 2 weeks (first 6 months), then monthly	Every 2 months	Every 3 months (if no laparoscopic second look) or at 12 months
II year	Every 2 months	Every 4 months	Every 2 months	Every 4 months	-
III year	Every 3 months	Every 6 months	Every 3 months	Every 6 months	-
IV year	Every 4 months	-	Every 4 months	Every 8 months	-
V–X year	Every 6 months	-	Every 6 months	Annualy	-

- means not scheduled.

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
