# Peer review of "Immature Teratoma of the Ovary—A Narrative Review"

_cancers, 2025, doi:10.3390/cancers17183041_

Round 1

Reviewer 1 Report

Comments and Suggestions for Authors

In this review article, the authors summarize the current evidence and recent advances in the diagnosis, prognostic factors, management, and follow-up of ovarian immature teratoma. This is a well-written and comprehensive review. I have only the following minor comments:

  1. Immature teratomas are a less common, but still reported, cause of anti-NMDAR encephalitis, compared to mature teratomas, which are more typically associated with this syndrome. I suggest the authors make some effort to address this interesting paraneoplastic phenomenon.
  2. Page 1, line 41: The statement “Epithelial tumors, arising from the coelomic epithelium lining the ovary” reflects an outdated concept. Current literature suggests that ovarian epithelial tumors may arise from the tubal epithelium, endometriosis, Brenner tumors, or teratomas. This should be clarified.
  3. Page 3, Table 2: “aFP” should be corrected to “AFP” for consistency with standard nomenclature.
  4. Page 4, Figure 2: It would be beneficial to include a corresponding H&E slide alongside the IHC image for better comparison and context.
  5. Page 10, lines 333 and 340: There appear to be some minor editing issues that need correction.

Author Response

Comment 1. Immature teratomas are a less common, but still reported, cause of anti-NMDAR encephalitis, compared to mature teratomas, which are more typically associated with this syndrome. I suggest the authors make some effort to address this interesting paraneoplastic phenomenon.

Thank you for the suggestion. Being so rare and less common with the immature counterpart of teratoma, we decided to not report the condition in this review. However, as you suggested, we’ve added a small paragraph dedicated to anti-NMDAR encephalitis.

Comment 2. Page 1, line 41: The statement “Epithelial tumors, arising from the coelomic epithelium lining the ovary” reflects an outdated concept. Current literature suggests that ovarian epithelial tumors may arise from the tubal epithelium, endometriosis, Brenner tumors, or teratomas. This should be clarified.

Thank you for your observation, we’ve updated the sentence

Comment 3. Page 3, Table 2: “aFP” should be corrected to “AFP” for consistency with standard nomenclature.

Thank you for your observation, we’ve corrected the term as you suggested

Comment 4. Page 4, Figure 2: It would be beneficial to include a corresponding H&E slide alongside the IHC image for better comparison and context.

Thank you, we’ve included the corresponding H&E slide (Figure 2a and 2b)

Comment 5. Page 10, lines 333 and 340: There appear to be some minor editing issues that need correction.

Thank you, we’ve edited some sentences on the text

Reviewer 2 Report

Comments and Suggestions for Authors

The manuscript addresses an important and relatively underexplored topic in gynecologic oncology. Immature teratomas (ITs) are rare, and a narrative review can be useful to both clinicians and researchers. The paper is well-structured and clearly written. The inclusion of tables summarizing previous studies and figures illustrating histological features is helpful for readers.

However, while the manuscript is overall solid, i have the following suggestions to improve the overall quality:

  • The manuscript claims that a comprehensive review of ITs is lacking. However, some previous reviews exist. The authors should better position their contribution, explicitly clarifying how their review differs (e.g., broader timespan, inclusion of fertility outcomes, synthesis of pediatric vs. adult data, or integration of molecular insights).
  • The “molecular and genetic” aspects of ITs are only briefly touched. Given the growing role of molecular profiling, the review should include more on genomic and epigenetic features, molecular differences with other germ cell tumors, and possible therapeutic implications.
  • The histological description is clear, but more discussion is needed on interobserver reproducibility of grading, pitfalls in frozen sections, and potential role of ancillary markers (e.g., SALL4, Glypican-3).
  • Differential diagnosis with other germ cell tumor and epithelial ovarian tumors should be discussed.
  • Additional histological figures should be included.

Author Response

Dear Reviewer #2, your contribution was very precious to improve and expand this review, covering all the aspects regarding immature teratomas.

Comment 1. The manuscript claims that a comprehensive review of ITs is lacking. However, some previous reviews exist. The authors should better position their contribution, explicitly clarifying how their review differs (e.g., broader timespan, inclusion of fertility outcomes, synthesis of pediatric vs. adult data, or integration of molecular insights).

Response 1. We’ve changed the sentence about the uniqueness of the review, clarifying the role of our manuscript and the existence of other reviews on specific topics and not a comprehensive one.

Comment 2. The “molecular and genetic” aspects of ITs are only briefly touched. Given the growing role of molecular profiling, the review should include more on genomic and epigenetic features, molecular differences with other germ cell tumors, and possible therapeutic implications. The histological description is clear, but more discussion is needed on interobserver reproducibility of grading, pitfalls in frozen sections, and potential role of ancillary markers (e.g., SALL4, Glypican-3). Differential diagnosis with other germ cell tumor and epithelial ovarian tumors should be discussed.

Response 2. We’ve broadened the molecular and genetic aspects including a new section in “diagnosis” (3.1 paragraph), paying also attention in including ancillary markers and the differential diagnosis with other ovarian tumors.

Comment 3. Additional histological figures should be included.

Response 3. As you suggested, we’ve also added a new figure (Figure 2b).
